# A Convolutional Neural Network for Beamforming and Image Reconstruction in Passive Cavitation Imaging

**DOI:** 10.3390/s23218760

**Published:** 2023-10-27

**Authors:** Hossein J. Sharahi, Christopher N. Acconcia, Matthew Li, Anne Martel, Kullervo Hynynen

**Affiliations:** 1Physical Sciences Platform, Sunnybrook Research Institute, Toronto, ON M4N 3M5, Canadaanne.martel@sunnybrook.ca (A.M.); 2Department of Medical Biophysics, University of Toronto, Toronto, ON M5G 1L7, Canada; 3Institute of Biomedical Engineering, University of Toronto, Toronto, ON M5S 3G9, Canada

**Keywords:** deep learning, beamforming, passive cavitation imaging, convolutional neural network, focused ultrasound

## Abstract

Convolutional neural networks (CNNs), initially developed for image processing applications, have recently received significant attention within the field of medical ultrasound imaging. In this study, passive cavitation imaging/mapping (PCI/PAM), which is used to map cavitation sources based on the correlation of signals across an array of receivers, is evaluated. Traditional reconstruction techniques in PCI, such as delay-and-sum, yield high spatial resolution at the cost of a substantial computational time. This results from the resource-intensive process of determining sensor weights for individual pixels in these methodologies. Consequently, the use of conventional algorithms for image reconstruction does not meet the speed requirements that are essential for real-time monitoring. Here, we show that a three-dimensional (3D) convolutional network can learn the image reconstruction algorithm for a 16×16 element matrix probe with a receive frequency ranging from 256 kHz up to 1.0 MHz. The network was trained and evaluated using simulated data representing point sources, resulting in the successful reconstruction of volumetric images with high sensitivity, especially for single isolated sources (100% in the test set). As the number of simultaneous sources increased, the network’s ability to detect weaker intensity sources diminished, although it always correctly identified the main lobe. Notably, however, network inference was remarkably fast, completing the task in approximately 178 s for a dataset comprising 650 frames of $41^{3}$ volume images with signal duration of 20μs. This processing speed is roughly thirty times faster than a parallelized implementation of the traditional time exposure acoustics algorithm on the same GPU device. This would open a new door for PCI application in the real-time monitoring of ultrasound ablation.

## 1. Introduction

The field of deep learning in medical ultrasound imaging is rapidly expanding with many applications such as clutter suppression in Doppler [1], super-resolution imaging [2,3], anomaly detection methods for breast ultrasound images [4], beamforming pre-steered, subsampled data [5], transcranial ultrasound imaging [6,7], minimum variance beamforming [8,9], and image reconstruction from raw channel data [10,11,12,13]. In the context of image reconstruction from raw channel data, CNNs have proven their capability to learn the reconstruction process without requiring explicit input information regarding receiver array geometry, a medium speed of sound, or the spatial discretization of the imaged region of interest (parameters which are typically essential for the standard delay and sum algorithm). This approach has been applied specifically to plane wave imaging [14,15,16,17] ultrasound molecular imaging [18], ultrasound B-mode imaging [19], ultrasound localization microscopy [20,21,22], and photo-acoustic imaging [23,24]. The potential application of deep learning in PCI has recently been explored, primarily through unsupervised learning using generative adversarial networks (GANs) as detailed in the work by Zeng et al. [25]. However, it is important to note that this approach was tested on simulated data involving a single source or bubble target. In practical scenarios, such as focused ultrasound treatments, it is important to investigate whether CNNs can effectively handle situations with multiple sources or bubbles. In another research study, a deep learning denoising approach is proposed based on a U-Net CNNs to reduce the image artifact [26]. Thus, the full extent of CNN’s adaptability to PCI remains an area of ongoing investigation.

PCI, an imaging method used for mapping cavitation sources, works based on the correlation of signals across an array of receivers. The greatest correlation of time-delayed signals determines the point of maximum beamformed intensity in the reconstruction grid. This technique, known as the ’time exposure acoustics’ (TEA) algorithm (Norton and Won 2000 [27]), has similarities with plane wave imaging [28], involving voxel-wise delay and sum operations along with an added time integration step. Currently, PCI is under investigation for real-time feedback control in the context of transcranial blood–brain barrier permeability (Jones et al., 2018 [29]). The control technique involves ramping up the driving pressure until bubble activity is detected based on the spatial coherence of bubble acoustic emissions and subsequently dropping the pressure incrementally. Thus, it is essential to generate images within the timeframes of therapeutic bursts (1 s) to identify acoustic sources. Since the image reconstruction rate is linked to the integration time, there is a keen interest in enhancing its speed, particularly for the extended burst durations (10 ms) used in therapeutic applications. To this end, the TEA algorithm can be implemented on GPU devices, resulting in significant performance enhancements, and ongoing optimization efforts are underway. Nevertheless, these classical PCI reconstruction methods, like TEA, need an intensive process of calculating the sensor weights for each pixel. As a result, employing these techniques for image reconstruction is not fast enough to meet the speed criteria necessary for real-time monitoring. An alternative approach could involve training a CNN to handle the image reconstruction process directly from raw channel data, and this paper explores the feasibility of such an approach.

CNNs are often employed to efficiently learn complex operations, as seen in the case of the minimum variance beamformer. However, in the case of TEA, the primary bottleneck is not the computational complexity but rather the number of global memory reads, the uncoalesced nature of these memory reads, and the number of parallel reduction operations performed for each voxel in the image. Consequently, one would not expect significant speed improvements from a CNN designed to mimic the delay, sum, and integration procedure (e.g., as in the approach of Mor and Bar-Hillel 2020 [10] to plane wave imaging). Nevertheless, with a suitable architecture, a CNN could potentially learn an efficient, alternative nonlinear mapping between raw channel data and the reconstructed image. This paper presents an evaluation of a CNN’s performance, trained to execute the PCI beamforming technique using simulated point source data. The primary objective of this evaluation was to assess the prediction accuracy, with a secondary focus on assessing the prediction speed. The outcomes of this study have the potential to create a novel opportunity for the use of PCI based on CNN in the real-time monitoring and controlling of ultrasound ablation procedures.

## 2. Materials and Methods

### 2.1. Passive Acoustic Monitoring

For over two decades, it has been demonstrated that low-intensity pulsed ultrasound can temporarily and selectively enhance the permeability of the blood–brain barrier (BBB) when microbubble contrast agents are present, all without causing damage to the surrounding brain tissue. This significant discovery, reported by Hynynen et al. [30], presents a promising approach for targets delivering therapeutic agents to the brain. However, the primary obstacle to the widespread clinical adoption of cavitation-based focused ultrasound (FUS) lies in the limitations of current techniques for the real-time monitoring and control of acoustic cavitation during treatment, as highlighted by Jones et al. [31].

The acoustic cavitation detection techniques are classified as active and passive acoustic monitoring. In the active detection approach, a focused high-frequency transducer, such as 30 MHz, emits a pulse during the transmit mode and captures the returning echo signal in the receive mode. Conversely, the passive detection method involves an unfocused, untuned transducer with a lower frequency, typically around 1 MHz, and it does not actively probe the cavitation zone [32,33].

Active acoustic monitoring comes with significant limitations. Since the intensity of therapeutic ultrasound signals is usually on a similar scale or much higher than those used for diagnostic purposes, it is necessary to deactivate the therapy beam while transmitting diagnostic pulses to prevent interference. Moreover, active techniques face the same challenges as transcranial acoustic imaging due to the aberration nature caused by the human skull bone, as noted by Jones et al. [31]. In contrast, passive acoustic monitoring has been frequently utilized to observe the acoustic emissions resulting from therapeutic exposures without involving the excitation of the region of interest. In this study, multi-element passive detector arrays were employed to spatially image the simulated cavitation activity.

### 2.2. Ray-Acoustic Model to Generate Radio-Frequency Data

A variety of numerical models have been developed for simulating the propagation of ultrasound waves. These models include ray-acoustics-based approaches [34,35], the angular spectrum method [36], full-wave methods [37], and hybrid models. In this research, a numerical model employing ray-acoustics has been utilized to simulate the propagation of ultrasound through water. The ray-acoustic models can accurately simulate high-frequency ultrasound waves, which is essential for medical ultrasound imaging [31,38]. Additionally, the ray-acoustic models are computationally efficient and suitable for simulating large-scale ultrasound wave propagation. They are often faster than full-wave methods [39].

Ray-acoustics is an approximation technique that calculates sound fields using the homogeneous, linearized wave equation of point-source response functions. In this method, discrete sound sources are subdivided into smaller sub-elements, and the resultant field is determined by summing the contributions from each sub-element. These sub-elements are subdivided to a degree where they can be accurately represented as point source emitters [31,34].

A small vibrating surface can be represented by an area of ds and a normal velocity of *u*, with its source strength indicated as uds. Assuming that the simple vibrating sub-element is positioned at coordinates $\left(x_{1},y_{1},z_{1}\right)$, because of its baffled harmonic radiation, we can express the acoustic pressure *p* at the point (x,y,z) as follows:(1)$$p\left(x,y,z\right)=\frac{jk_{c_{w}}\rho_{w}c_{w}}{2\pi }\cdot \frac{e^{-jk_{c_{w}}R}}{R}\cdot \left(uds\right)$$

In this case, the longitudinal particle velocity can be expressed as [31]:(2)$$u\left(x,y,z\right)=\frac{jk_{c_{w}}}{2\pi }\cdot \frac{e^{-jk_{c_{w}}R}}{R}\cdot \left(uds\right)\cdot \left(1-\frac{j}{k_{c_{w}}R}\right)$$

In Equations (1) and (2), $k_{c_{w}}$ is a complex wave number that can be calculated as:(3)$$k_{c_{w}}=\omega /c_{w}-j\alpha_{w}$$
where ω=2πf is the angular frequency of the sub-element, *j* is the imaginary unit ($j^{2}=-1$), $\alpha_{w}$ is the attenuation coefficient, $\rho_{w}$ and $c_{w}$ are the density and sound speed of the medium, respectively. The variable *R* corresponds to the distance between the source coordinates $\left(x_{1},y_{1},z_{1}\right)$ and the point of the receiver (x,y,z). Equation (Equation 1) presents an alternative expression of the classical Rayleigh–Sommerfeld integral, specifically for a single simple source [40].

### 2.3. Time Exposure Acoustics for Image Reconstruction

The acoustic images, often referred to as "ground-truth" images, were generated using the time exposure acoustics (TEA) algorithm, which is detailed in Norton and Won [27] (2000) and Norton et al. [41] (2006). This algorithm is closely related to the conventional delay-and-sum (DAS) beamforming technique [42,43].

To generate an image using the TEA method, a set of control points within a ROI, where image reconstruction is to be carried out, needs to be defined. To create the beamformed signal at each control point, the signals received from the detector array are independently scaled, delayed, and summed together [31]. Then, an image is constructed by calculating intensity per each voxel using the temporal mean of the beamformed signal intensity (Norton and Won, 2000 [27]). In this study, receive beamforming was executed in the time domain, following the approach outlined by Gyongy and Coussios [44].

The image intensity within an integration window ranging from $t_{o}$ to $t_{f}$ is calculated as follows:(4)$$I\left(r\right)=\int_{t_{o}}^{t_{f}}\left[\sum_{i=1}^{N}Q_{i}{\left(r;t\right)}^{2}\right]\;dt\;$$
where $Q_{i}$ is the time delayed signal for the receiver number *i* at the reconstruction point *r*:(5)$$Q_{i}\left(r;t\right)=p_{i}\left(\frac{|R_{i}|}{c_{w}}+t\right)\cdot \left|R_{i}\right|$$

Here, $p_{i}$ is the signal received by array element *i*, delayed by the time of flight in water $\left(|R_{i}|/c_{w}\right)$, $|R_{i}|$ is the magnitude of the vector between the receiver and the reconstruction point *r*, and $c_{w}$ is the speed of sound in water. The beamforming process was carried out with an integration time of 20 µs. For the reconstructions, a 3D region of interest with dimensions of 10×10×10 cubic millimeters ($0.25^{3}$ mm voxel size) was selected. The center of this ROI was positioned at distances of 2 cm, 4 cm, and 6 cm away from the center of the receiver array. Figure 1 shows the 2D array receivers probe (16×16) which were located 4 cm away from the 3D ROI. Table 1 provides details regarding the ultrasound parameters and material properties used for simulating the ray-acoustics model for radio-frequency (RF) generation, along with the parameters employed for TEA beamforming. The speed of sound and attenuation properties of water were determined using the values reported in Duck [45] and Pajek and Hynynen [34].

### 2.4. Convolutional Network Architecture for Image Reconstruction

In this study, we employed a 3D CNN-based approach to reconstruct passive acoustic images from raw RF signals. The architecture of the 3D CNN was adapted from a 2D CNN employed by Anas et al. in their 2018 work on photoacoustic imaging [13].

In essence, this network consists of a series of dense blocks, each with an increasing size of dilated convolutions. In this approach, as illustrated in Figure 2 and detailed in Table 2, we introduced strided convolution layers in the time domain for downsampling and a transposed convolution layer in the spatial domain for upsampling between these dense blocks. This step-by-step process gradually reshaped the input data with dimensions (1521, 16, 16) into the desired image dimensions (41, 41, 41). Within the dense blocks, batch normalization layers were incorporated, and for the skip connections, we used the addition of layer outputs rather than concatenation, as depicted in Figure 2.

The network comprises a total of 21 convolutional layers, and it contains 150,317 learnable parameters. The rectified linear unit (ReLU) activation function was applied to all layer outputs except for the final layer. It is worth noting that the output of the final layer was normalized, as our primary interest lay in the relative differences in voxel values rather than in their absolute values.

## 3. Simulation and CNN Input Materials

Simulation data were employed to train, validate, and test the proposed network. The training data were generated with point sources randomly positioned within a 3D volume that covered the entire volumetric reconstruction grid. The number of point sources per reconstruction varied between 1 and 5. Each source had the same amplitude and was modeled as a continuous wave, monochromatic emitter with a separate set of frequencies including 256 kHz, 612 kHz, and 1.0 MHz. The ray acoustics model was utilized to propagate these point sources to a 16 × 16 matrix probe of receivers. The simulation and data training were carried out on three distinct datasets. The first dataset used a source frequency of 612 kHz, a receiver element size of 2.48 mm, and a distance of 4 cm between the source’s center and the probe. The second dataset employed a source frequency of 256 kHz, a receiver element size of 5.8 mm, and a distance of 6 cm between the source’s center and the probe. The third dataset involved a source frequency of 1000 kHz, a receiver element size of 1.5 mm, and a distance of 2 cm between the source’s center and the probe. It’s noteworthy that in all these datasets, the receiver element size, center-to-center spacing, was set to a wavelength of the ultrasound frequency. A separate set of network weights have been employed for different dataset.

To enhance the network’s robustness against noise, white noise with a Gaussian distribution was introduced into the channel data. Simulated cavitation signals were generated at a 40 MHz sampling rate, and noise was added to achieve a mean signal-to-noise ratio (SNR) of −4 dB across all channels. The standard deviation across different receive elements was set to 0.3 〈SNR〉, which was based on typical variations observed in experimental data, as reported by Acconcia et al. in 2017 [46].

Before beamforming using the TEA method, the received signals were digitally filtered using a sixth-order Butterworth filter with bandpass frequencies centered around the source frequency, with a half-bandwidth of 300 kHz. All the generated images were divided into three groups to form the training, validation, and test sets. These images were distributed independently, with 2000 images for training, 25 for validation, and 500 for testing.

### 3.1. Network Evaluation and Optimization

To prepare the input data for the network, the raw RF channel data (without time delays) underwent temporal trimming to retain only the essential information. This involved selecting time points ranging from the minimum delay among all receivers to the maximum delay among all receivers, plus the integration time. Consequently, when beamforming the trimmed data using the traditional algorithm, relative time delays between the elements were used instead of absolute delays. Following trimming, during the preprocessing stage, each input was z-score normalized.

The target consisted of 3D beamformed volumes generated by the standard TEA algorithm. In the preprocessing stage, these target volumes were normalized and converted into pressure values (i.e., the square root of the beamformed intensity). This conversion aimed to distribute the L1 loss function’s weighting more evenly between the side lobe regions and main lobe regions.

Various image quality metrics are commonly used to assess the quality of images, including normalized root mean squared error (NRMSE), peak signal-to-noise ratio (PSNR), and structural similarity index measure (SSIM). In this study, we specifically focused on evaluating the suitability of SSIM, NRMSE, and PSNR for the task at hand.

Following the approach of Simson et al. in 2018 [5], we utilized a loss function that incorporated both the SSIM loss and the $l_{1}$ loss. The $l_{1}$ loss function, also known as the mean absolute error (MAE), is used to measure the absolute differences between predicted and actual values. The formula for the $l_{1}$ loss can be expressed as [5]:(6)$$l_{1}\left(I_{tru},I_{pred}\right)=\frac{1}{N}\sum_{i=1}^{N}\left|I_{tru}\left(i\right)-I_{pred}\left(i\right)\right|$$
where $I_{tru}$ and $I_{pred}$ (both sizes of *N*) indicate the intensity values in ground truth and predicted images, respectively.

To assess the model’s accuracy, a custom version of SSIM was employed, using a 3D Gaussian kernel to calculate variance at each point. The formula for Custom SSIM is:(7)$$SSIM\left(I_{tru},I_{pred}\right)=\frac{\left(2\mu_{tru}\mu_{pred}+C_{1}\right)\left(2\sigma_{tru/pred}+C_{2}\right)}{\left(\mu_{tru}^{2}+\mu_{pred}^{2}+C_{1}\right)\left(\sigma_{tru}^{2}+\sigma_{pred}^{2}+C_{2}\right)}$$
where $\mu_{tru}$ and $\mu_{pred}$ represent the means of $I_{tru}$ and $I_{pred}$, respectively; $\sigma_{tru}$ and $\sigma_{pred}$ are the standard deviations of $I_{tru}$ and $I_{pred}$, $\sigma_{tru/pred}$ is the covariance between $I_{tru}$ and $I_{pred}$, and $C_{1}$ and $C_{2}$ are small constants to stabilize the division with weak denominators.

It is important to note that all these values are in tensor form, and the overall SSIM was calculated by taking the mean across all cells.

In terms of the relative weighting of the two loss functions, we placed a higher emphasis on the SSIM loss, attributing it to 84% of the total loss. It is worth mentioning that the L1 loss function, when used in isolation, proved insufficient for overfitting even when applied to a small subset of the training data. To leverage the local 3D statistical information within the target volumes, we implemented a 3D version of SSIM using TensorFlow’s basic operations. This utilized a Gaussian window with a size of 11 voxels and a standard deviation of 1.5 voxels.

Furthermore, the PSNR, which is calculated in decibels (dB) based on the mean square losses between the estimated and reference images, is defined as:(8)$$PSNR=20\log_{10}\frac{I_{max}}{\sqrt{MSE}}.$$
where
(9)$$MSE=\frac{1}{N}\sum_{i=1}^{N}{\left(I_{tru}\left(i\right)-I_{pred}\left(i\right)\right)}^{2}$$
where $I_{max}$ represents the maximum intensity in the reference image.

As another image quality metric, NRMSE was also used to measure the differences between the predicted images and ground truth images. It quantified the Euclidean distance between the point of maximum intensity in the target image and the predicted image, representing the source localization error. The NRMSE formula is defined as:(10)$$NRMSE=\frac{\sqrt{MSE}}{I_{max}-I_{min}}$$
where $I_{max}$ and $I_{min}$ represent the maximum and minimum intensities in the reference image.

To optimize the convolutional network, the loss function was run with a range of learning rates. The learning rate is a hyperparameter that determines the step size at which the model parameters are updated during training. In the context of the Adam optimizer, the learning rate is one of the most important hyperparameters. It controls the size of the steps taken to minimize the loss function. A larger learning rate can lead to faster convergence but may risk overshooting the optimal solution, while a smaller learning rate can provide a more stable convergence but might take longer to reach the optimal solution. In this study, an optimized learning rate of 0.0003 was selected for training the network using the learning rate scheduler in TensorFlow.

The network was trained on an NVIDIA GeForce RTX 2070 Super GPU using the TensorFlow backend. We employed the Adam optimizer [47] and applied L2 regularization (0.001) for training based on the training set. A batch size of five was used, and the model was trained for 300 epochs.

### 3.2. Passive Acoustic Imaging Analysis and Statistical Testing

For each reconstruction, the image quality and the number of sources detected were compared between the target and the predictions. The peak sidelobe ratio (PSLR) was defined as the ratio between the maximum intensity of the side lobes and the main lobe’s intensity. Furthermore, the SNR was characterized as the ratio of the maximum intensity within the reconstruction to the standard deviation of the background signal (i.e., all voxels greater than a wavelength from the main lobe’s maximum intensity, as detailed in Jones et al. [48]).

The determination of the number of sources present was accomplished by applying a threshold at the −3 dB level and subsequently identifying the count of contiguous volumes within the binarized volume. It is important to note that the number of sources detected in the reconstruction might not necessarily align with the number of point sources present in the simulations. This discrepancy could arise from the potential separation distance between point sources being smaller than the point spread function (PSF) of the receiver array. To analyze these differences, confusion matrices were generated to compare the number of sources detected in the target data with those in the network reconstructions.

In order to assess the similarity in image quality between the target and predicted groups, we conducted a pair-wise sign test specifically on the image SNR and PSLR metrics.

## 4. Results

In this section, the performance of CNNs have been presented for a particular network weight trained for a source frequency of 612 kHz, a receiver size of 2.48 mm, and a 4 cm distance between the probe and the ROI location. Additional research and simulations have demonstrated that the CNNs’ performance remains relatively stable across a range of frequencies, receiver sizes, and ROI locations as detailed in Table 1.

The training and validation loss curves, as shown in Figure 3, do not exhibit clear indications of overfitting. Qualitatively, it can be observed that the network has effectively learned the underlying beamforming process, as evidenced by the agreement between the 3D contour plots and 2D cross-section images of the ground truth and the network predictions (refer to Figure 4 and Figure 5). Moreover, the network has demonstrated its capability to accurately reconstruct not only the main lobe region but also the diffraction pattern of the receiver array in the lower intensity regions (as shown in Figure 6 and Figure 7).

As shown in Figure 6 and Figure 7, there is a discrepancy between the quantity of sources detected in the reconstruction and the number of point sources simulated. This inconsistency, as mentioned earlier, could be attributed to the potential scenario where the separation distance between the point sources is smaller than the PSF of the receiver array, or it may also result from significant variations in the strengths of the sources.

To perform a quantitative assessment of the CNN methods, we calculated SSIM, RMSE, and PSNR indices for varying numbers of sources within the target volume. As shown in Table 3, the results show that the highest prediction accuracy is achieved when there is a single isolated source. Detecting and monitoring a single isolated source is crucial for the real-time localization of therapy targets in focused ultrasound. Therefore, this CNN model can serve as a valuable predictive tool for determining the maximum pressure and intensity levels during ultrasound therapy. All evaluation metrics in Table 3 are computed using the testing dataset.

Quantitatively, this is supported by a pair-wise sign test on the target and predicted PSLR values, which indicated that the median difference between the two was not significantly different from zero (p>0.05). In the case of the image SNR, the same test revealed a significant difference (p<0.05) between the target and predicted values, although the median and interquartile range (IQR) of the pair-wise difference were small (−0.4, IQR = 2).

For the intended clinical application, the ability to detect and locate individual sources is of the primary focus. In all instances where single isolated sources were identified in the ground truth volumes, the network made accurate predictions 97% of the time (a summary is provided in Table 4). As the number of sources detected in the target volume increased, the true positive rate (TPR) decreased (as shown in Table 4) to approximately 45%. Nevertheless, it is noteworthy that, in general, the network deviated by just one or two sources and consistently identified the largest volume source or the source with the maximum intensity.

The network predictions also exhibited a strong performance in terms of spatial localization accuracy and main lobe size estimates. The median error in the predicted maximum intensity voxel location was 0.5 mm, with an interquartile range (IQR) of 0.8 mm. Regarding the error in the main lobe radius, the median and IQR values were 0.02 mm (IQR = 0.05 mm), 0.01 mm (IQR = 0.05 mm), and 0.3 mm (IQR = 0.6 mm) for the short axes and the long axis, respectively.

In terms of inference time, for the detection of five sources at a distance of 2 cm away from the receiver array, processing a batch of five volumes took an average of 1.37 s ± 6 ms, as measured by the tensorboard profiler. This duration includes the overhead time associated with kernel launches and data transfer between the host and the device.

It is important to emphasize that the choice of loss function significantly impacted the network’s performance. Solely employing the L1 loss function resulted in poor network training, even when attempting to overfit a small-sized training set. This can be attributed to the more balanced weighting of the loss function across both high- and low-intensity regions provided by the SSIM component. When using a pure L1 loss function, the relatively small number of high-intensity voxels becomes the primary contributor to error backpropagation during weight parameter updates, potentially leading to a loss of valuable training information.

## 5. Discussion

This paper demonstrated the CNN’s potential to effectively learn the TEA beamforming algorithm. The results indicate that the network can accurately predict the key features of volumetric images, including image SNR, PSLR, main lobe size, and location. Additionally, it can capture more complex features, particularly in the low-intensity regions away from the main lobe.

The network accurately predicted the exact number of sources present in reconstructions when the number of sources detected was low. For instance, it achieves a true positive rate (TPR) of 97% and a false positive rate (FPR) of only 0.02% for single-source images. This low FPR is crucial from a safety perspective, preventing erroneous pressure amplitude increases between bursts in the therapeutic controller schemes. However, as the number of detected sources in the target increases, the TPR for correctly identifying the number of sources decreases, reaching approximately 45% for cases with five sources. It is important to note that the network still identifies local intensity maxima at the locations of these missed sources, and the only reason they were not counted was that their local maximum values fell below the −3 dB threshold. The decrease in TPR with increasing source numbers could be attributed to potential bias in the training data. If multiple point sources are simulated with a separation distance smaller than the PSF of the receiver array, they will not be resolved as separate sources in the target reconstructions. Consequently, while equal amounts of training data were generated for each number of point sources, cases with distinct sources in the beamformed volume are fewer, potentially introducing bias in favor of detecting fewer sources.

The CNN’s reconstruction speed was found to be quite fast, approximately 178 s for a dataset of $650\times 41^{3}$ volume images. This speed is approximately thirty times faster than a parallelized implementation of the TEA algorithm on the same GPU device used for network inference. It is worth noting that no specific optimization for inference speed, such as network pruning or exploiting 16-bit floating-point operations with tensor cores, was attempted in this study, which primarily aimed to investigate the network’s feasibility for image reconstruction. That being said, there is also potential room for improvement through the further optimization of the CUDA implementation, so this comparison should be made with caution.

An interesting observation in Figure 3 is the sharp increase in accuracy at around 100 epochs. While several factors could contribute to this phenomenon, one speculative reason is that the model has not yet adequately learned all "types" of data until that point. Once it has learned the desired patterns, accuracy sharply improves as predictions align with more stable patterns.

This paper introduced an initial investigation of employing a CNN for PCI beamforming. However, it’s important to note that the chosen approach has few limitations that worth further discussion. Firstly, the performance evaluation was conducted solely on simulated data. However, some efforts were made to incorporate the variability one might encounter experimentally, such as variable receiver sensitivity and noise profiles. The noise profiles were randomly assigned to receivers for each simulation, meaning that the sensitivity of a particular receiver was not constant across the dataset. The network’s performance on experimental data as input is a subject of ongoing research. Secondly, the training data only involve continuous wave sources, whereas experimental studies can exhibit time-dependent behaviors during long therapeutic bursts or may be transient in nature. Future investigations will assess the network’s ability to handle finite-duration sources and distinguish between the sources of variable durations. Thirdly, the receiver center-to-center spacing in this study was set to a wavelength of the ultrasound frequency, which is suitable for imaging along the center axis of the array without grating lobes. The grating lobes could become prominent for source locations and reconstruction points far away from the main axis (i.e., for large steering angles). In this study, the imaging locations were situated in front of the array within a relatively small volume, which prevented grating lobes from entering the imaging field. In cases with large fields, smaller receivers in fully populate form with a center-to-center spacing of half a wavelength or randomly populated arrays may be required to avoid grating lobes. Lastly, some specific clinical applications (e.g., Jones et al., 2018 [31]) may involve sparsely populated, hemispherical receiver arrays, which would necessitate the design of a different network architecture. Improved networks, possibly employing graph convolutional networks (GNNs) [49,50], or extreme learning machine (ELM) [51,52], may be needed to account for the added complexity of the receiver geometry and non-uniform receiver positioning. The similar challenges have been encountered in cosmology (Perraudin et al., 2019) and climate prediction, where observations are made over an approximately spherical geometry and at unevenly separated locations on the surface. These advancements will constitute a focal point for our forthcoming research.

## 6. Conclusions

This study has demonstrated the effective application of a CNN in implementing the time exposure acoustics algorithm for PCI. By incorporating a custom loss function consisting of L1 and SSIM components, the network achieved the highest SSIM and PSNR scores across a dataset of 2000 simulations. The spatial localization of sources exhibited a high degree of accuracy, with a median error and interquartile range of 0.5 mm and 0.8 mm, respectively. It exhibited the robustness to noise and sensitivity in identifying image reconstructions with isolated single sources. These findings indicate that the use of 3D-CNN in simulated ultrasound acquisition can lead to faster acquisitions without compromising the image quality. This technique holds promise for medical applications, particularly in cavitation-based-focused ultrasound, where rapid acquisition and reduced processing times are critical. Future research will focus on assessing the performance of 3D-CNN using experimental data in a more comprehensive clinical study aimed at the real-time monitoring and control of acoustic cavitation.

## Figures and Tables

**Figure 1 sensors-23-08760-f001:**
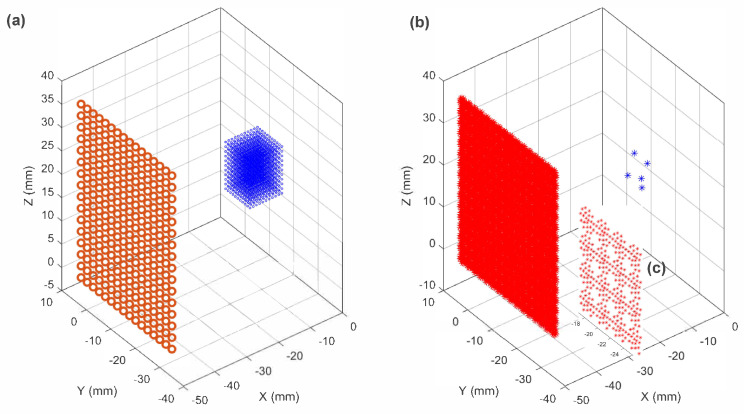
(**a**) The 3D region of interest (ROI) had dimensions of 10×10×10 m^3^ and was positioned at a distance of 4 cm from the center of 2D receiver probes with 16×16 elements. A separate set of network weights have been employed for different ROI locations. (**b**) Five (5) sources are randomly located in 10×10×10 m^3^ for one dataset (**c**) Zoom-in window on receiver probes which are discretized and subdivided into smaller sub-elements.

**Figure 2 sensors-23-08760-f002:**
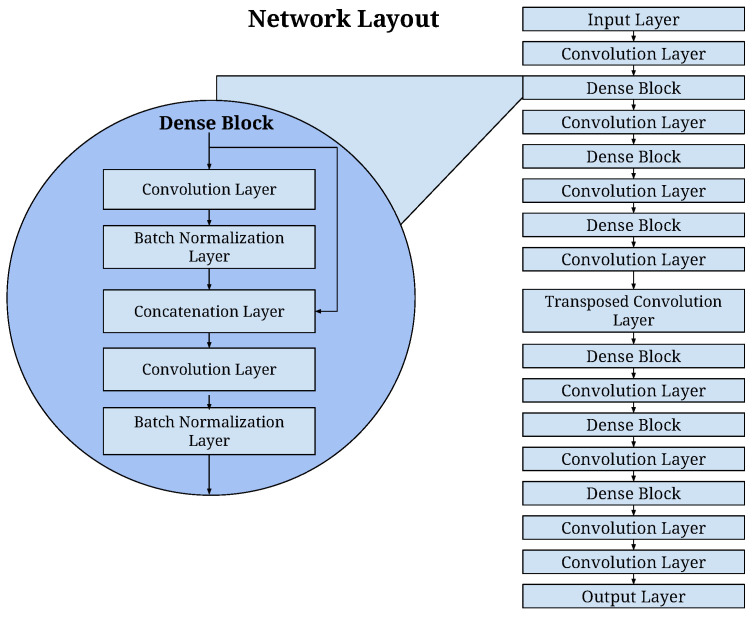
The CNN architecture to map the channel data to a passive acoustic image. The network consists of six (6) dense blocks, where each dense block includes two densely connected convolutional layers followed by batch normalization layer. The difference among six (6) dense blocks lies in the amount of dilation factor.

**Figure 3 sensors-23-08760-f003:**
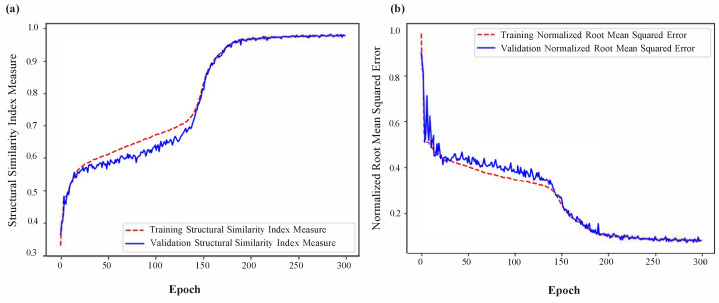
(**a**) Structural similarity index measure vs. epoch; and (**b**) normalized root mean squared error vs. epoch.

**Figure 4 sensors-23-08760-f004:**
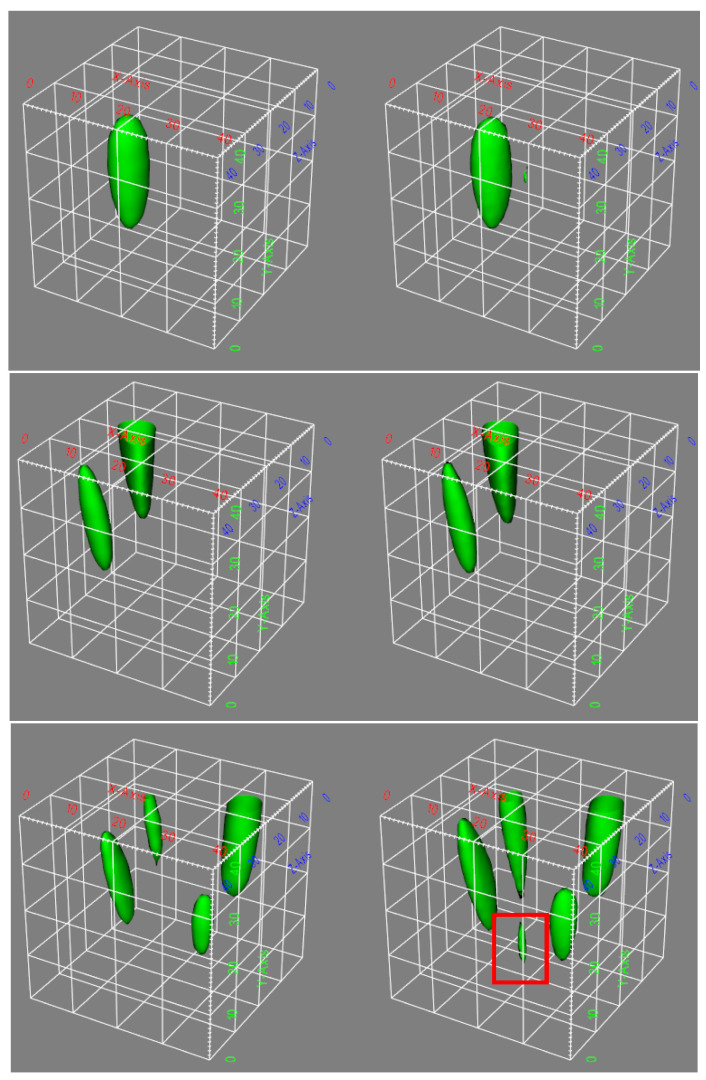
An example of simulated passive acoustic images with varying numbers of sources. On the left are the ground truth reconstructions, and on the right are the predictions made by the network. One can observe that the network exhibits accurate predictions when the number of sources is less than three. However, for cases with four sources, the prediction accuracy decreases. The red rectangle highlights the disparities between the ground truth and the network’s predictions in the case of four sources. The X, Y, and Z axes represent the pixel counts for image reconstruction.

**Figure 5 sensors-23-08760-f005:**
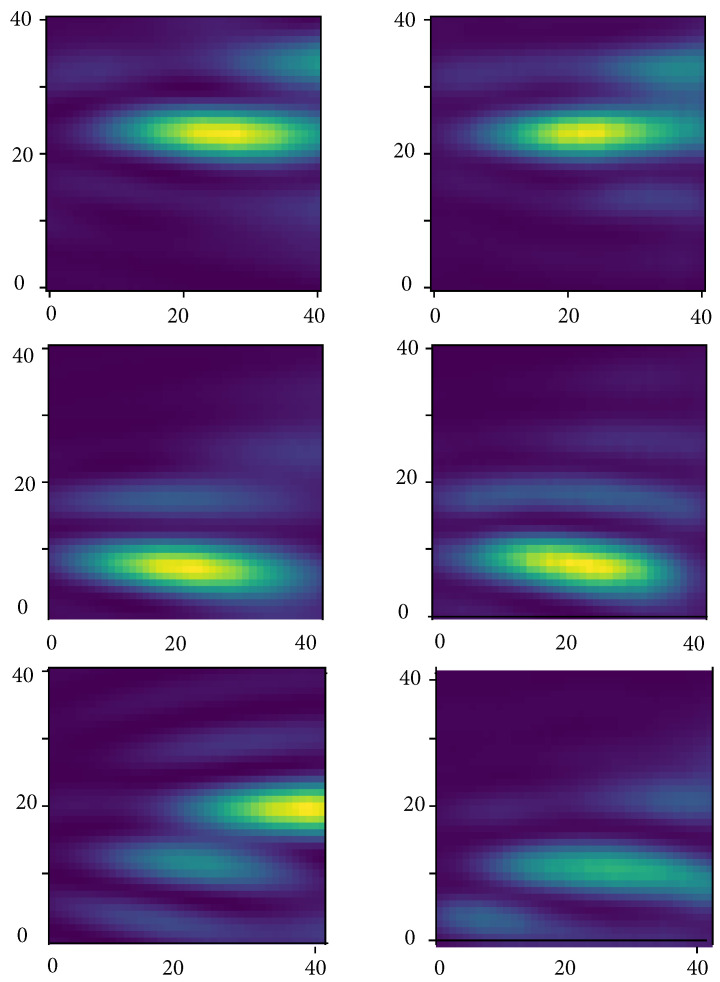
On the left side is a cross-section of the reconstructed data with *beamforming*. On the right side, there is the same cross-section of the reconstructed data using the neural network.

**Figure 6 sensors-23-08760-f006:**
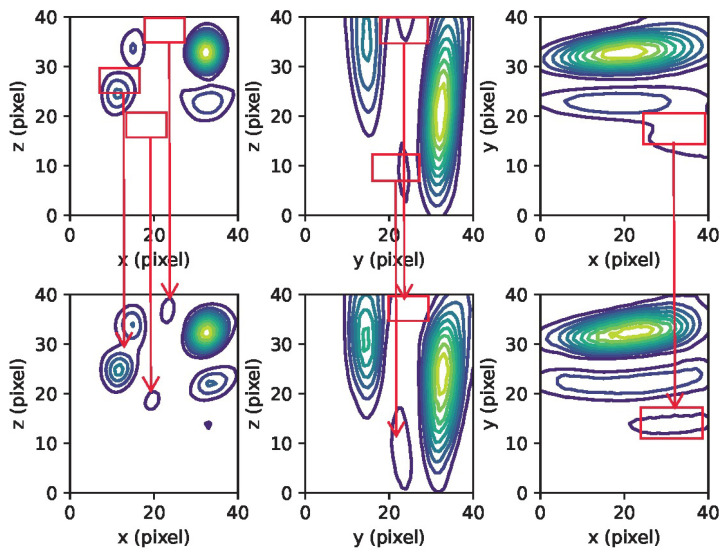
Contour plots displaying the maximum intensity projection for a case with three (3) sources (linear contours at 10% intervals) are presented. The top section shows the ground truth, while the bottom section displays the CNN’s prediction. The red rectangles emphasize the differences between the ground truth and the network’s prediction for the three sources.

**Figure 7 sensors-23-08760-f007:**
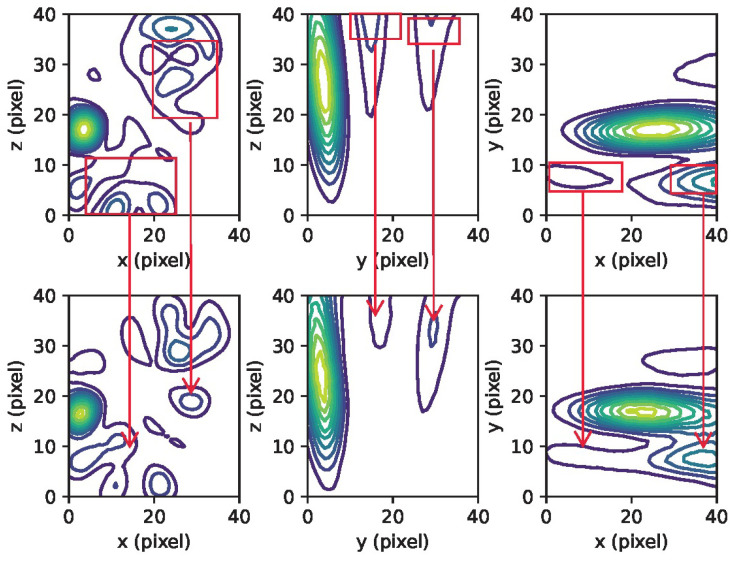
Contour plots displaying the maximum intensity projection for a case with five (5) sources (linear contours at 10% intervals) are presented. The top section shows the ground truth, while the bottom section displays the CNN’s prediction. The red rectangles emphasize the differences between the ground truth and the network’s prediction for the three sources.

**Table 1 sensors-23-08760-t001:** The parameters used in the simulation of the ray-acoustic model, along with the parameters employed for TEA beamforming.

Description	Parameter	Value	Unit
Receiver frequency	$f_{o}$	256, 612, 1000	kHz
Attenuation of sound in water	α	$4.32\times 10^{-4}$	Np/m
Density of water	ρ	1000	kg/m^3^
Speed of sound in water	$c_{w}$	1500	m/s
Size of receiver probe	*d*	1.5, 5.8, 2.48	mm
Sampling rate	$f_{s}$	40	MHz
Signal-to-noise ratio	SNR	−4	dB
Standard deviation of SNR	$\sigma_{SNR}$	0.3	–
Region of interest dimensions	$ROI_{l}$	10×10×10	mm^3^
Number of receiver probes	$NRx_{y}$ × $NRx_{z}$	16×16	–
Distance between center of source and probe	$L_{ROI-Probe}$	2, 4, 6	cm
Number of points per wavelength for discretization	ppwl	4	–
Length of signal in time	$Rx_{Duration}$	140	μs
Number of sample in one duration	$Rx_{samples}$	140×40=5600	–

**Table 2 sensors-23-08760-t002:** The output shape, dilation, and kernel size details of the convolutional network architecture.

Layer Type	Output Dimension	Kernel Size	Dilation	Stride	Number of Kernels
Input shape	1521, 16, 16	–	–	–	–
Convolutional layer	1521, 16, 16	9	–	1	2
Dense block (1)	1521, 16, 16	9, 3, 3	1	1	2
Convolutional layer	761, 16, 16	9	–	2, 1, 1	4
Dense block (2)	761, 16, 16	9, 3, 3	2	1	4
Convolutional layer	381, 16, 16	9	–	2, 1, 1	8
Dense block (3)	381, 16, 16	9, 3, 3	4	1	8
Convolutional layer	191, 16, 16	5	–	2, 1, 1	8
Convolutional transpose layer	191, 48, 48	6	–	1, 3, 3	8
Dense block (4)	191, 48, 48	9, 3, 3	4	1	8
Convolutional layer	96, 48, 48	5	–	2, 1, 1	8
Dense block (5)	96, 48, 48	9, 3, 3	4	1	8
Convolutional layer	48, 48, 48	8	–	2, 1, 1	8
Dense block (6)	48, 48, 48	9, 3, 3	4	1	8
Convolutional layer	44, 44, 44	5	–	1	8
Convolutional layer	41, 41, 41	4	–	1	8
Output shape	41, 41, 41	–	–	–	–

**Table 3 sensors-23-08760-t003:** Quantitative evaluation of the proposed CNN method for different numbers of sources.

Number of Sources	SSIM ^1^	PSNR ^1^	RMSE ^1^
1	0.978	39.4	0.019
2	0.867	38.7	0.091
3	0.812	32.3	0.192
4	0.750	25.3	0.228
5	0.425	17.8	0.336

^1^ SSIM: structural similarity index measure; PSNR: peak signal-to-noise ratio; RMSE: root mean squared error.

**Table 4 sensors-23-08760-t004:** Confusion matrix values for correctly predicting the presence of isolated sources.

Number of Sources	TPR ^1^	FPR ^1^	TNR ^1^	FNR ^1^
1	0.97	0.02	0.96	0.03
2	0.88	0.09	0.92	0.12
3	0.82	0.06	0.91	0.18
4	0.70	0.06	0.91	0.25
5	0.45	0.02	0.95	0.45

^1^ TPR: true positive rate; FPR: false positive rate; TNR: true negative rate; FNR: false negative rate.

## Data Availability

The corresponding author can be contacted for access to the data presented in this study upon request. The data are not publicly available due to restrictions on privacy.

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
