# Peer review of "A Convolutional Neural Network for Beamforming and Image Reconstruction in Passive Cavitation Imaging"

_sensors, 2023, doi:10.3390/s23218760_

Round 1

Reviewer 1 Report

This manuscript focused on evaluating the performance of CNN for the PCI beamforming, the topic looks interesting. More details please refer to the following comments:

1) Both the motivations and contributions are not clear in Abstract and Introduction, please refine them.

2) The full names of all the abbreviations should be given when they appear for the first time, such as SNR and RF. .

3) Separate related work section should be considered by the authors to review state-of-the-arts are in this area.

4) High-quality figures are strongly suggested to better demonstrate both the proposed method the experimental results.

5) The experiments are not sufficient. More baseline methods, evaluation metrics, scenarios and datasets should be included to support the proposed method.

6) The authors should comprehensively discuss both the limitations of the proposed method and future directions. For example, how to further enhance the performance of the proposed method using deep learning with prior knowledge. Some related papers are recommended, which are better included in the reference list: physics-informed deep learning for musculoskeletal modeling: predicting muscle forces and joint kinematics from surface emg, IEEE TNSRE, and non-iterative and fast deep learning: multilayer extreme learning machines, JFI.

English writing improvement should be considered by the authors.

Reviewer 2 Report

English language editing suggested. Poor mathematical interpretation. More equations relevant to the work expected and explain the novelty of your work justifying mathematical modelling. Results need to be explained with more in-depth analysis. Comparison with state of art techniques not given. Is the proposed solution the best approach? Is the proposed solution clearly articulated so that it reflects that this is a well thought plan with clearly defined steps for solving the proposed problem? 10 Relevant references from 2023 need to be added in the literature survey. 

Revise the paper accordingly and resubmit.

Use grammarly for getting information regarding all errors.

Reviewer 3 Report

It's an interesting topic for me to review. Is there any clinical application sample for your study?

Thank you and best regards! 

NA

Round 2

Reviewer 1 Report

The authors have addressed the issuses, I have no more comments.

The authors have addressed the issuses, I have no more comments.